Extending beyond individual caves: a graph theory approach broadening conservation priorities in Amazon iron ore caves

http://orcid.org/0000-0002-2174-1970 Oliveira Marcus P. A. 1 marcuspbr@gmail.com
http://orcid.org/0000-0003-3288-4405 Ferreira Rodrigo L. 2
1 BioEspeleo Consultoria Ambiental , Lavras, MG , Brazil
2 Center of Studies in Subterranean Biology, Ecology and Conservation Departament, Universidade Federal de Lavras , Lavras, MG , Brazil
Root-Bernstein Meredith
Electronic publication date: 2024 Jan 31
Publication date: 2024
Volume: 12
Electronic Location ID: e16877
Received 2023 Oct 9; Accepted 2024 Jan 11
Copyright: © 2024 Oliveira and Ferreira
Copyright year: 2024
Copyright holder: Oliveira and Ferreira
License: This is an open access article distributed under the terms of the Creative Commons Attribution License, which permits unrestricted use, distribution, reproduction and adaptation in any medium and for any purpose provided that it is properly attributed. For attribution, the original author(s), title, publication source (PeerJ) and either DOI or URL of the article must be cited.
License URL: https://creativecommons.org/licenses/by/4.0/

Keywords: Cave connectivity, Conservation, Communities, Troglobites, Amazonian biodiversity

Funding: Convênio de cooperação técnica e tecnológica between Vale Institute of Technology-Sustainable Development and Universidade Federal de Lavras, N° 189/2014 This study was funded by Convênio de cooperação técnica e tecnológica between Vale Institute of Technology-Sustainable Development and Universidade Federal de Lavras, N° 189/2014. The funders had no role in study design, data collection and analysis, decision to publish, or preparation of the manuscript.

==============================
The Amazon is renowned worldwide for its biological significance, but it also harbors substantial mineral reserves. Among these, the ferruginous geosystems of the region are critical for iron ore extraction, accounting for 10% of Brazil’s export revenue. Additionally, this region holds a significant speleological heritage with more than 1,000 caves. However, cave conservation efforts are often in conflict with land use, necessitating mediation through environmental regulations. While conservation decisions typically consider only the caves’ characteristics, such an approach fails to account for the interactions among cave communities and their surrounding landscape. This poses a challenge to reserve design for cave conservation purposes. To address this issue, we assessed the predictors that influence the similarity among cave communities, suggesting the use of this parameter as a proxy for subterranean connectivity. Applying graph theory, we proposed a tool to aid in the selection of priority caves for conservation purposes. Our study involved the sampling of invertebrates in 69 iron ore caves and analyzing 28 environmental variables related to these subterranean habitats and adjacent landscape. Our analysis revealed that landscape and habitat characteristics are more important than geographical distance in determining patterns of similarity among caves. Our graph approach highlighted densely interconnected clusters based on similarity. However, specific caves stood out for harboring exclusive fauna and/or exhibiting habitat specificity, making them unique in the study area. Thus, we recommend prioritizing cave clusters for conservation, assembling both singular caves and others that influence them. It is crucial to note that protocols for the protection of subterranean biodiversity must consider measures that encompass both the caves and the surrounding landscape. Our methodology provides insights into the connectivity among caves, identifies existing groups, highlights singular (or unique) cavities that require preservation, and recognizes those influencing these unique habitats. This methodological advancement is crucial for the development of better conservation policies for the speleological heritage in areas under constant economic pressure.

Introduction

The Amazon region is renowned for its extraordinary biological diversity (Oliveira et al., 2017), sheltering the largest tropical rainforest in the world (Cardoso et al., 2017). In addition to its biological importance, this region harbors significant mineral reserves, including open-pit mines for iron, nickel, copper, and gold extraction, which generate 10% of Brazil’s exports (Piló, Auler & Martins, 2015; Agência Nacional de Mineração–ANM, 2018). Notably, the Amazonian ferruginous geosystems contain over 1,000 recorded caves, despite their typically small size (approx. 30 m long) (Piló, Auler & Martins, 2015). These caves host highly diverse invertebrate communities, including many cave-restricted species (troglobites) (Jaffé et al., 2016, 2018; Ferreira, Oliveira & Silva, 2018). The relative shallowness of these caves, coupled with an intricate network of interconnected void spaces within the rock matrix, enables nutrient exchange and migration of species between surface and subterranean habitats (Ferreira, Oliveira & Silva, 2018). Despite their ecological significance, the preservation of these habitats is often challenged by mineral exploitation. The preservation of a single cave in this region can represent the withholding of around US $38 billion in commodity, given the ferruginous matrix in which they are embedded (Auler & Piló, 2015).

Conflicts between cave conservation and land use are not restricted to the Amazon region. Karst areas are widely distributed around the world and have been affected by urban and agricultural expansion in almost all the continents. In addition, the increasing demand for minerals, many of which are associated with caves, poses a threat to these habitats (Ferreira, Oliveira & Silva, 2015; LaMoreaux, Powell & LeGrand, 1997; Ferreira et al., 2022). Therefore, many countries established specific laws to mediate such conflicts, such as USA (National Historic Preservation Act (NHPA), 1966; Archaeological Resources Protection Act-ARPA, 1979; Native American Graves Protection & Repatriation Act-NAGPRA, 1990), China (Zhifang, 1991; People’s Republic of China, 1989), Australia (Environmental Protection Authority-EPA, 2016, 2007), Slovenia (The Republic of Slovenia, 2003), Slovakia (The Republic of Slovakia, 2002) and Brazil (Federal Republic of Brazil, 2008; Auler & Piló, 2015). However, for all of these countries, current conservation decisions regarding caves often consider only their intrinsic attributes, while ignoring the importance of adjacent landscapes and interactions among different caves and other karst compartments (Christman et al., 2016; Jaffé et al., 2018; Mammola et al., 2020). This impasse raises questions about the most effective conservation strategies: whether to preserve individual caves or groupings of interconnected caves.

The discussion on nature reserve design began in the 1970s, focusing on whether a single large or several small areas should be used for species conservation purposes (referred to as SLOSS) (Simberloff & Abele, 1976; Ovaskainen, 2002; Groenveld, 2005; Tjørve, 2010). This debate has guided conservation proposals in various ecosystems, including tropical forests (Laurance & Bierregaard, 1997), marine environments (McNeill & Fairweather, 1993), and even urban areas (Valente, Pasimeni & Petrosillo, 2020), as well as for different groups of species, such as butterflies (Baz & Garcia-Boyero, 1996) and birds (Lindenmayer et al., 2015). However, this approach has not yet been applied to subterranean fauna. As a result, important caves are often isolated in fragmented landscapes, limiting the dispersal of species, including troglobitic species (Jaffé et al., 2018; Hoch & Ferreira, 2012). Therefore, how can we define conservation targets considering a scale larger than the cave itself? One possibility is to use biological parameters of cave communities associated with landscape and habitat information, such as similarity matrices, to support an assessment among caves rather than only for each cave individually.

In this study, we aimed to identify the predictors that influence the similarity among cave communities, proposing the use of this analysis as a proxy to preserve the subterranean fauna associated with Amazon ferruginous geosystems. Specifically, we sought to answer the following questions: (a) Does the similarity between cave communities differ when considering the entire community compared to only troglobitic species? (b) To what extent can we detect such similarities? (c) Which factors are more influential in shaping these patterns: geographical proximity, habitat similarity, or landscape traits? (d) How can we leverage similarity to identify connectivity between caves, and in this context, (e) what are the most suitable conservation strategies? As hypotheses, we expect to find differences in the similarity between caves depending on the species considered in the analysis (troglobitic vs non-troglobitic), due to differences in the degree of endemism for each of these groups. We expect that the proximity between caves will outweigh habitat and landscape traits in determining similarity, considering the increased likelihood of species migration between nearby communities. We anticipate identifying clusters of nearby caves with similar troglobitic species due to their subterranean interconnectivity. Finally, we predict that strategies focused on the conservation of cave clusters will be prioritized over individual preservation efforts.

Materials and Methods

Study site

We carried out this study in 69 iron-ore caves located in Serra do Tarzan (6°19′32″S, 50°06′36″W), eastern region of the Amazon biome, Pará state, Brazil (Fig. 1). Serra do Tarzan is situated within the Campos Ferruginosos National Park, which is part of a network of protected areas known as the Carajas Mosaic, surrounding the mineral extraction province (Carajás Mineral Province). We selected this study area because it represents the closest unit to the pristine condition of the Amazon ferruginous geosystem. Geologically, this region is characterized by banded iron formation (BIF), represented by jaspilites exhibiting intercalations of iron oxide and silica. The hills are covered with a thick ferruginous lateritic layer, known as canga, where caves develop, preferably at the edges of plateau fractures (Piló, Auler & Martins, 2015). The climate is characterized by an average temperature ranging from 23 °C to 25 °C and an annual rainfall of approximately 2,400 mm (Piló, Auler & Martins, 2015). However, the distribution of rainfall is irregular, with the rainy season occurring between October and April (with a range of 160 to 340 mm per month), and a dry period spanning from May to September (with a range of 10 to 90 mm per month) (Sahoo et al., 2016).

Figure 1 Location of the 69 cavities assessed in the present study.

In the detail: Serra do Tarzan location in Eastern Amazon, Southeastern Pará state, Brazil. Map data © 2018 Google/Maxar Technologies.

Sampling cave fauna and variables

We sampled each cave twice in 2016: first during the rainy season (between January and February) and subsequently during the dry season (between July and August). During fieldwork, we measured 28 different variables to characterize the caves (grouped into physical features, lithology, trophic characteristics, climatic factors) and their surrounding landscape (location, cave insertion, climate, vegetation) (Tables S1–S3). Additionally, we sampled the arthropod fauna throughout the cave area with the involvement of at least three speleologists using active capture (sampling effort: 6 min per square meter per collector). Permission was given by Instituto Chico Mendes de Conservação da Biodiversidade-ICMBio, N° 83/2016. All the caves were thoroughly inspected, as they are relatively small and have limited food resources, resulting in a lower abundance of fauna. We fixed all invertebrates in 100% ethanol and sent them to specialists for identification to the lowest possible taxonomic level (see Acknowledgments). We identified the specimens as morpho-species, an approach commonly used in ecological biodiversity studies and conservation purposes (Oliver & Beattie, 1996; Pellegrini et al., 2016; Oliveira et al., 2019). Troglobites were recognised based on their troglomorphic traits, as validated by each specialist responsible for the identifications (Howarth & Moldovan, 2018). Vouchers of each taxon were deposited in reference collections (institution names and accession numbers are available in Dataset S1). The full datasets can be found along with the R scripts (Dataset S1).

Data analysis

Similarity assessment among cave communities

The assessment of community similarity within cave ecosystems was carried out employing two distinct methodologies: (i) an examination of species composition, and (ii) an analysis of phylogenetic distance. The former approach involved comparing the similarity of species identities across communities, treating each cave as an individual community. Conversely, the latter approach investigated whether these communities consisted of species with comparable evolutionary relationships, as indicated by phylogenetic distance. Both approaches were applied to all species present within the communities, as well as exclusively to the troglobitic species, facilitating the differentiation of similarity patterns influenced by surface environments from those occurring exclusively in subterranean habitats.

We conducted a principal coordinate analysis (PCoA) utilizing the Bray-Curtis dissimilarity index to compare the overall species composition among caves, as well as specifically for troglobitic species. These analyses were performed using the “capscale” function from the “vegan” R package, employing a matrix with standardized and log-transformed abundance data (log(x + 1), where “x” represents the number of individuals observed for each species). For subsequent analyses, we utilized the first two axes of the PCoA (MDS1 and MDS2) as response variables, which served as proxies for the general species composition and troglobic species composition, respectively.

The expected phylogenetic distance between taxa from different communities was determined using the mean pairwise distance (MPD) metric. We carefully selected 411 invertebrate taxa from 37 Orders that were widely distributed throughout the study area to represent the entire species set (see Table S4). Our selection criteria focused on taxonomic accuracy, ensuring that all morphotypes were identified at least up to the genus level. To validate our approach, we conducted a Spearman’s correlation analysis, which confirmed a strong and significant relationship between the selected taxa (SPHY) and the total number of species (STOT) present in the cave communities of the study area (SPHY vs STOT: rho = 0.970 and p < 0.001).

Regarding troglobites, we considered all taxa in this category. To assess the phylogenetic distance, we transformed the hierarchical taxonomic identity into a cluster representation that quantified the distance between taxonomic levels. We normalized these distances by the number of levels, following the method proposed by Clarke & Warwick (1999). Subsequently, we utilized the comdist function from the “picante” R package to calculate the MPD values, which provided the distances between each pair of communities. We visually represented these results using Principal Coordinate Analysis (PCoA) with the capscale function from the “vegan” R package, employing the Euclidean distance metric. Similar to the analysis of species composition, we utilized the first two axes of the PCoA (MDS1 and MDS2) as proxies for phylogenetic distance in subsequent models, both for all species and exclusively for troglobitic taxa.

Spatial autocorrelation in cave communities

To examine the influence of geographical distance on the similarity among cave communities, we employed Moran’s I spatial autocorrelation statistic. This analysis was conducted on the response variables of species composition and phylogenetic distance, considering both all species and only troglobitic species. The correlog function from the “pgirmess” package was utilized for this purpose. The results were presented in correlograms, where spatial autocorrelation values ranged from −1 (indicating perfect dispersion) to +1 (indicating perfect correlation), with values near 0 representing random distribution. Significance of the indices (p-values) for each distance class was assessed using progressive Bonferroni correction, following the approach proposed by Legendre & Legendre (2012).

Modeling similarity patterns among cave communities

To investigate the factors contributing to the observed spatial autocorrelation, we employed a distance-based linear model (DistLM) in cases where significant spatial autocorrelation was detected. This analysis aimed to determine whether geographic proximity, habitat similarity, and/or external landscape traits were responsible for the observed pattern. The DistLM analysis allowed us to assess the contribution of each predictor to the similarity approach considered for the analyzed species (all species and/or troglobitic species). We conducted these analyses using the PRIMER 7 software with the PERMANOVA extension (Anderson, Gorley & Clarke, 2008).

We used the 28 variables obtained from the caves and external landscape as predictors. Categorical variables were transformed into binary variables, with ‘1’ indicating the presence of a category and ‘0’ indicating its absence, following the approach described by Anderson, Gorley & Clarke (2008). These binary variables were then grouped into sets representing their respective categories using the Indicators function. In this step, we grouped the geographical coordinates (UTM East and North) in a single set to represent the distance (in meters) among caves.

We chose adjust R2 as selection criterion and Forward as selection procedure, which starts with the null model and adds one variable at a time choosing at each step the variable that results in the greatest improvement in the selection criterion value (Anderson, Gorley & Clarke, 2008). Thus, we identified the significant models (those with p < 0.050 after the addition of new predictors) for the observed spatial autocorrelation pattern and their respective explanatory power.

Delimiting thresholds for cave community similarity

We employed piecewise regression analysis on the significant variables identified by the DistLM to detect trend changes (breakpoints) concerning the response variables. This analysis was conducted using the “segmented” R package. The percentage of similarity between pairs of caves was adopted as response variable for species composition (for both the whole community and for troglobitic species). These similarity values were calculated using the Bray-Curtis index, as it provides more informative results in the context of our study. The vegdist function from the “vegan” R package was utilized for this calculation.

Regarding phylogenetic distance, we used mean pairwise distance (MPD) values between pairs of communities. Similarly, we transformed the predictor variables into pairwise differences using Euclidean distance matrices. We applied piecewise regression using generalized linear models (Gaussian distributed errors with logistic link function for response variables). We compared AICc values between the types of regression (linear or by parts) to select the best application, as indicated in Ochoa-Quintero et al. (2015) and Magnago et al. (2015). Results were expressed considering the amplitude of variation (Δ) of the segmented model in relation to the linear one. Finally, we applied orthogonal contrast tests for categorical predictors to individualize the significant ones (glht function from “multcomp” R package).

Applying graph theory to determine caves connection and conservation priorities

Troglobic species are unable to establish viable populations in epigean environments, as highlighted by Culver & Pipan (2014). Consequently, their dispersal is limited to underground habitats, primarily occurring through the canaliculi found in ferruginous geosystems, as demonstrated in studies by Ferreira, Oliveira & Silva (2015, 2018), Jaffé et al. (2016, 2018), and Trevelin et al. (2019). Therefore, when the same troglobic species is observed in multiple caves, it can indicate a connection between these habitats through the ferruginous matrix.

Connectivity analysis typically incorporates the area of sampling units, the distances between them, species’ dispersal capabilities, and immigration rates as key factors (Hanski, Alho & Moilanen, 2000; Martensen, Pimentel & Metzger, 2008). Accordingly, we used the predictor values derived from spatial autocorrelation analysis and piecewise regression (breakpoints) for troglobitic species’ similarity as indicators of cave proximity and habitat characteristics within the sample units. The similarity in species composition allowed us to identify groups of caves sharing common troglobites, while phylogenetic similarity provided insights into their dispersal potential, given that closely related species often exhibit similar dispersal capabilities (Webb et al., 2002; Gaston, 2003; Beutel et al., 2017). Finally, since we considered all the species in the group (troglobites) and all their occurrence locations (assuming that individuals from the same population are restricted to the whole study area), the immigration rate is absent, since all movement possibilities for specimens were considered in the model (as indicated in Rösch et al., 2013).

We employed graph theory, following the methodology outlined by Gross & Yellen (1999), to assess the connections between cave communities. Graphs were constructed and analyzed using the “igraph” R package. Initially, we created four graphs to represent the potential pairwise connectivity (CPC) between caves based on different similarity measures (species composition and phylogeny) and analysis techniques (spatial autocorrelation and piecewise regression). In the case of spatial autocorrelation, CPC was considered when the distance between caves was equal to or smaller than the distances identified as significant in the Moran’s I correlogram with progressive Bonferroni correction. Regarding piecewise regressions, CPC was determined based on the values of significant predictors that favored similarity, as indicated by the breakpoints.

We constructed graphs to represent the strength of connections by assigning weights to the potential pairwise connectivity (CPC) based on the four previously mentioned possibilities. In this process, we validated connections that exhibited at least one metric of composition and phylogeny, as they indicate the “identity” and dispersal capability of species among caves, respectively. We assigned three different weights: Weight 2) CPC indicated by two graphs; Weight 3) CPC indicated by three graphs and Weight 4) CPC indicated by all the graphs. The weight of connections between caves corresponded to the level of similarity among troglobitic species, thus indicating the degree of connectivity among sets of caves. Finally, we created a parsimonious graph that represented the connections between caves based on the established weights.

We conducted a comparison of pairwise connectivity (CPC) by assessing the density of connections across all the graphs. The density provides an indication of the proportion of existing connections among all possible connections (calculated using the edge_density function). Additionally, we employed the Girvan & Newman (2002) approach to create clusters based on edge betweenness in order to identify groups of caves that exhibit higher connectivity (utilizing the cluster_edge_betweenness function). To further evaluate the connectivity patterns, we applied the modularity function to determine whether there are more connections within groups of caves compared to connections with caves outside the group. Noteworthy connections, serving as “bridges” between distant regions of the graph, were identified using the intermediation centrality measure (betweenness function).

To characterize the uniqueness of each cave in relation to its troglobitic fauna and habitat specificity, we defined a singularity measure. Singularity (SC) was calculated using the formula: SCΩ = (STRΩ/STRTOT)/(DΩ/DMAX), where: STRΩ = number of troglobic species in the cave Ω; STRTOT = total number of troglobitic species for the study area; DΩ = number of connections observed for the cave Ω (obtained by the degree function); DMAX = maximum number of different possible connections between caves in the study area.

Connections between caves with and without troglobitic species were included in all graphs, as caves lacking troglobites exhibited a combination of predictors that favored the presence of this group of organisms, as indicated by previous analyses such as spatial autocorrelation, DistLM, and piecewise regression. We acknowledged that the absence of troglobic species in a particular cave could be attributed to sampling limitations (false negatives). However, we excluded potential connections between caves where no troglobitic taxa were found, as the absence of such species may suggest the absence of suitable habitat conditions for their occurrence.

Considering the graph-based approach, we identified priority caves for conservation based on their singularity and connection strength (weight). Caves with Sc > 1 were designated as having the highest priority for preserving local diversity. These caves exhibited a higher proportion of troglobitic taxa compared to their connections, indicating their habitat specificity within the study area. Subsequently, for each high-priority cave, we assessed its connections with other caves to determine the communities that influenced them. The level of influence was determined based on the decreasing strength of connections, with those exhibiting the highest CPC Weight among the connections established with the priority conservation cave classified as “high” influence, followed by “medium” and “low”. Thus, we identified secondary priority caves that exerted a high influence on the high-priority caves with a significant presence of troglobitic species. Specific graphs were developed for each conservation priority cave, illustrating the connections and levels of influence from other cave communities.

Results

Cave fauna and variables

The analysis of 28 environmental variables allowed us to differentiate caves based on their habitat and landscape characteristics (see Table S5). Our extensive sampling effort resulted in the collection of 163,092 specimens, comprising 693 species, 245 families, and 51 orders (refer to Dataset A1). Among these species, we identified nine as troglobitic (refer to Fig. S1).

Cave community similarity

We found that the similarity values between caves were higher when considering only the troglobitic fauna, compared to when including all species (see Figs. S2A and S2B). In terms of the phylogenetic approach, the clusters revealed that certain diverse classes, such as Arachnida and Insecta, had limited representation among troglobitic species (see Fig. S3). This suggests that while cave communities may exhibit phylogenetic dissimilarity in relation to the overall fauna, they can still display taxonomic similarity specifically among troglobites (see Figs. S2C and S2D).

Spatial autocorrelation for cave communities

The species composition and the presence of phylogenetically related taxa exhibited notable similarities among cave communities located within a radius of 700 m from a given cave (see Figs. 2A and 2B). When considering the cave-restricted fauna, we observed a significant influence on species composition for caves located up to 1,243 m apart (see Fig. 2C). Furthermore, the presence of troglobitic species in a particular community promoted the occurrence of phylogenetically related taxa in neighboring caves situated within a distance of up to 745 m (see Fig. 2D).

Figure 2 Spatial autocorrelation for the similarity approaches (species composition or phylogenetic distance) applied to all the species or only troglobites.

Indices range from −1 to 1 in the correlogram. Positive values indicate positive spatial association between the places (correlation), while negative values indicate negative association (dispersal). Values around zero indicate random distribution without spatial association. The red filled circle () represents significant correlations and the empty one (○) the non-significant (p > 0.050). the grey filled circle () indicates non-significant correlations after Bonferroni progressive correction.

Similarity patterns among cave communities

Although neighboring caves can be influenced by the composition and presence of phylogenetically related species, geographical proximity does not always result in similar communities (see Table 1). In fact, the distance between caves primarily contributes to the phylogenetic similarity and the presence of shared troglobitic species when combined with other predictive factors (see Table 1). Conversely, species composition is more strongly influenced by landscape factors, such as forest cover, scarp height, temperature, as well as habitat traits like humidity and hydric features (see Table 1). Our findings also reveal that the occurrence of phylogenetically related troglobites is influenced by physical, trophic, and water-related traits, alongside the geographic position and climate of the surrounding landscape (see Table 1).

Table 1 Relationship of similarity metrics with landscape and cave features.

Similarity approach	Predictor variables	Adjust R2	Pseudo-F	p	
Species composition	Hydric features	0.037	3.588	0.002	
Forest cover	0.073	3.627	0.001	
Minimum humidity	0.094	2.552	0.001	
Scarp height	0.112	2.309	0.003	
Mean temperature	0.122	1.734	0.015	
Troglobites composition	Distance	0.173	6.548	0.004	
Scarp height	0.330	12.917	0.002	
Maximum humidity	0.487	16.348	0.001	
Phylogenetic distance (All Species)	Distance	0.003	1.108	0.002	
Area	0.005	1.146	0.006	
Minimum humidity	0.007	1.092	0.018	
Phylogenetic distance (Troglobites)	Area	0.120	8.251	0.002	
Granulometry	0.196	2.634	0.009	
Mean temperature	0.244	4.115	0.002	
Bat guano	0.267	2.524	0.039	
Maximum humidity	0.285	2.183	0.059	
Altitude	0.301	1.997	0.065	
Scarp height	0.315	1.922	0.083	
Hydric features	0.334	2.292	0.036	

Thresholds for cave community similarity

We identified specific ranges of values that contribute to the similarity between pairs of caves (see Fig. 3), taking into account the significant predictors identified in the models (see Table 1). For the overall fauna composition, caves exhibited greater similarity when their relative humidity values were within a range of ±26% RH, escarpment heights were similar (±11 m), water features had a similar seasonal pattern, caves were surrounded by forests within a range of ±10 ha, and the average epigean temperature was approximately ±0.1 °C (see Fig. 3A). In terms of phylogenetic distance, we observed greater similarity between communities in caves that were geographically closer (±618 m), had a similar area (±191 m2), and exhibited a minimum humidity level within a range of ±18% RH (see Fig. 3B).

Figure 3 Relationship between the similarity approaches applied to all the species or only troglobites and respective significant predictors.

The graphs display thresholds for significant predictors of similarity in: (A) Species composition (all species). (B) Phylogenetic distance (all species). (C) Troglobites composition. (D) Phylogenetic distance (all species). In graphs of segmented regression the dashed vertical line (in blue) indicates the significant breakpoint (p < 0.050), the grey area represents the standard error and ΔAICc denotes the variation of such model AIC in relation to the linear one. In boxplots, blue areas refer to the confidence interval (95%) around the observed average (central black line) and bars represent the standard deviation. In all the graphs light pink dots represent the sampling units (caves).

Regarding the troglobitic species composition, we observed similarities between caves that are located until 5 km apart, inserted in escarpments with slight differences in height (±3.5 m) and that present relative air humidity with similar values (MaxH ±4% RH) (Fig. 3C). On the other hand, caves hosting communities with phylogenetically related troglobites displayed similarities in area (±185 m²), relative humidity (MaxH ±6% RH) and guano accumulation (±0.25 m²). Furthermore, these caves shared similar water features, and the predominant sediment granulometry differed by at most one level (e.g., if one cave had a predominance of sand, the others would have a predominance of granules or silt) (Fig. 3D). Additionally, regarding the surrounding landscape, phylogenetically similar communities were observed in caves located at similar altimetric levels (±70 m), cliffs (±8.5 m), and experiencing similar epigean average temperatures (±0.6 °C) (Fig. 3D).

Cave connections and conservation priorities

We found that the pairwise connectivity (CPC) was higher when considering the species composition compared to the phylogenetic distance (see Fig. 4). Additionally, the graph generated from spatial autocorrelation exhibited higher CPC values, in contrast to what was observed for the phylogenetic distance graph (see Figs. 4A and 4B).

Figure 4 Connections between pairs of caves (CPC) according to metrics applied to the similarity approaches for troglobites.

CPC value represents density of connections for each graph (indication of the proportion of existing connections among all possible connections). (A) CPC based on species composition similarity trough spatial autocorrelation. (B) CPC based on species composition similarity trough piecewise regressions. (C) CPC based on phylogenetic distance trough spatial autocorrelation. (D) CPC based on phylogenetic distance similarity trough piecewise regressions.

All the studied caves are connected when considering both composition and phylogeny traits (connections of Weight 2) (Fig. S4). We found 27 caves with at least one connection that encompasses all the similarity metrics used (Weight 4) (Fig. S4). Based on the most parsimonious graph, we observed that 21% of all potential connections among caves in Serra do Tarzan were present (see interactive 3D graph in Fig. S5). The cluster analysis revealed the presence of six groups of caves that are densely interconnected (Modularity = 0.472) (Fig. 5). Regarding the intermediation centrality, eight caves acted as bridges between different regions in the study area: ST_0038, ST_0039B, ST_0014, ST_0027, ST_0018, ST_0063, ST_0039A and ST_ 0012 (see Table S6). Finally, we indicate 16 caves as priorities for conservation since they present a singularity value higher than one (Fig. 6 and Table S6). These caves are highly influenced by 23 other distinct caves, which we recommend as secondary conservation priorities (Fig. 6).

Figure 5 Groups of densely connected cavities.

(A) Hierarchical tree showing the community structure for the network calculated by edge betweenness. (B) Spatial distribution of groups and respective caves of Serra do Tarzan, highlighting connections among them.

Figure 6 Influence relationships for caves indicated as maximum priority for conservation.

Communities with high influence (red edge) over each priority (purple vertex) cave are highlighted.

Discussion

Our findings provide support for the hypothesis that the similarity among caves varies depending on whether all species or only troglobitic ones are considered. Interestingly, the spatial range of similarity among caves based on troglobitic species was larger than expected. Surprisingly, we found that geographical distance does not play the most crucial role in determining patterns of similarity among caves. Instead, landscape and habitat characteristics emerged as more influential factors, as evidenced by their significant predictors across all similarity measures examined. Our predictions of similar troglobitic communities within clusters of nearby caves were confirmed, underscoring the existence of subterranean connectivity between them. In terms of conservation, our hypothesis was partially supported: it is important to prioritize cave clusters for conservation, but these clusters should consist of both unique caves and those capable of influencing biological communities. We believe that this approach strikes a better balance between cluster conservation and individualized recommendations.

We identified variations in similarity metrics (species composition and phylogenetic distances) when considering the entire cave community vs only troglobitic species. Caves serve as ecotones bridging surface and subterranean ecosystems (Moseley, 2009), providing refuge for species from different landscape compartments (de Fraga et al., 2023). As a result, pairs of caves can exhibit either similarities or distinctiveness depending on the specific set of species considered. Interestingly, we found that the taxonomic groups with the highest diversity in the overall cave community were not well represented among troglobites. Instead, it was the troglobitic species that contributed most significantly to the similarities observed among caves. This implies that cave communities diverge as we include species associated with other compartments, such as surface ecosystems, many of which are locally restricted and unrelated to troglobites.

We also observed a narrower spatial range for similarity measures when considering the entire community species. The ferruginous geosystems encompass a diverse range of landscapes, ranging from savannahs to forests (Mota et al., 2015). Therefore, non-troglobitic populations found within the caves are distributed based on the local conditions and resources available in both the cave and surface ecosystems. Consequently, the similarity between distant cave communities is influenced by euryecic species that can adapt to various conditions or by the level of homogeneity in the external landscape components. In the latter case, the troglobitic components of the communities achieve a broader range as the subterranean habitats they inhabit attenuate variations in external environmental conditions. Likewise, the narrower spatial range of phylogenetic similarity, in comparison to species composition, reflects the distinct tolerance capabilities of each taxonomic group in adapting to environmental conditions, thereby influencing their dispersion patterns according to specific characteristics and limitations.

Jaffé et al. (2018) emphasized the role of geographic distance as the primary factor influencing community similarity in Amazon iron ore caves. However, we underscore the significance of landscape and habitat characteristics, which can also influence similarity measures and may overlap with the importance of geographic distance. The divergent results obtained in our study compared to the aforementioned study appear to be associated with the predictor characteristics employed. While Jaffé et al. (2018) considered physical and trophic characteristics of caves, their landscape data was obtained at a larger scale. In contrast, our study utilized a wide range of fine-scale predictors, providing a higher resolution of conditions and resources both within the caves and in the surrounding landscapes. Moreover, our research was conducted in a pristine environment, whereas the previous study examined caves located near areas of mineral exploitation and livestock (Jaffé et al., 2018). It is worth noting that the authors of the previous study suggested that anthropogenic land use did not necessarily act as a barrier to subterranean connectivity (Jaffé et al., 2018). However, such changes might have fragmented the landscape, limiting species dispersal to groups of closely located caves, thus amplifying the importance of geographic distance in determining community similarity.

We observed that landscape characteristics had a predominant influence on species composition, whereas habitat attributes emerged as determinants of phylogenetic similarity. This suggests that the landscape plays a role in selecting the species occurring across the entire study area, while only groups already pre-adapted to subterranean conditions thrive within the caves. Howarth & Hoch (2012) reported a higher likelihood of subterranean colonization for taxa whose preferred habitats correspond to those found in caves.

Regarding troglobitic species, there are physical (e.g., area, granulometry), climatic (e.g., water features, humidity), and trophic (e.g., guano) constraints that influence the phylogenetic composition similarity between cave pairs, but there are no spatial limitations. This indicates that for many taxa, the ferruginous matrix itself comprises the preferred habitat, and these species are found in caves only when specific favorable conditions are present. Studies conducted in ferruginous areas of Australia have demonstrated a high diversity of taxonomic groups in shallow subterranean habitats (Eberhard et al., 2009; Guzik et al., 2011; Halse & Pearson, 2014; Halse, 2018). In Brazil, the majority of the known fauna in ferruginous ecosystems has been found in caves, as most studies have focused on these habitats (Ferreira, Oliveira & Silva, 2018). Therefore, expanding research to consider different compartments of the ferruginous landscape may reveal new patterns of phylogenetic similarity between different regions within such systems.

Troglobitic species exhibit lower tolerance to variations in habitat and landscape traits compared to non-troglobitic species. This reaffirms the notion of the extensive distribution of troglobites within the ferruginous matrix (Ferreira, Oliveira & Silva, 2018; Jaffé et al., 2016, 2018). Many troglobitic species are exclusive to habitats with specific environmental conditions. Moreover, it highlights the vulnerability of troglobitic populations to even slight changes in certain environmental variables, which can lead to their local extinction. We emphasize once again the role of anthropogenic activities, such as mining and livestock farming, in selecting which troglobitic species persist in the landscape. The elimination of certain taxa enhances the similarity among cave communities. Consequently, the high similarity observed among pairs of caves in disturbed landscapes (as indicated by Jaffé et al., 2018) may be attributed to the homogenization of the remaining fauna, favoring the more resilient troglobitic species and artificially increasing connectivity among cave communities.

We observed distinct values for the similarity measures, wherein geographic distance emerged as a significant predictor when considering spatial autocorrelation or segmented regression. Through the application of the Moran’s I test, we were able to assess the degree to which the similarity measure in a particular cave affects the occurrence of similar values in neighboring habitats. The breakpoint obtained from the segmented regression provided insight into the distance at which the relationship with the similarity between cave pairs becomes statistically insignificant. Hence, we propose the combined use of these two approaches as complementary measures for landscape zoning. The intersection area resulting from the application of both distance metrics to a given cave indicates a higher level of influence among cave communities. Non-congruent regions that still encompass one of the distance metrics represent areas with relatively lower potential to influence the inter-cave fauna interactions.

Although several studies have discussed the connectivity among ferruginous caves (Ferreira, 2005; Souza-Silva, Martins & Ferreira, 2011; Jaffé et al., 2016, 2018; Ferreira, Oliveira & Silva, 2018), this study presents the first empirical model that elucidates such connectivity. Our observations reveal that this connectivity is primarily determined by the specific identity of individual taxa rather than the broader taxonomic group to which they belong. The graphs obtained for the different similarity measures and approaches indicate that widely distributed species exhibit phylogenetic relatedness, whereas spatially restricted species belong to distinct taxonomic groups. Consequently, the greater phylogenetic diversity is confined to nearby caves that possess unique combinations of habitat characteristics, which are also influenced by the limited dispersal capability of certain species.

Despite the observed connectivity among all caves in the study area, we identified specific clusters of caves that exhibited a higher degree of interconnection. This finding raises important considerations regarding current environmental legislation, particularly in countries like Brazil (Auler & Piló, 2015), where caves are evaluated and managed individually. By solely focusing on habitat attributes during conservation prioritization, there is a risk of promoting landscape fragmentation, leading to the gradual isolation of cave communities. Under such circumstances, arthropod communities are likely to experience a reduction in diversity (α, β, ψ) and altered spatial-temporal dynamics, ultimately impacting ecosystem functioning (Bestion et al., 2019). To ensure the preservation of interactions among cave communities and the sustainable utilization of speleological heritage in the face of mineral and/or agricultural exploitation, it is crucial to adopt a holistic approach that considers the connectivity and interdependence of cave systems.

Even within clusters that exhibit high connectivity based on similarity, certain caves stand out for harboring an exclusive fauna and/or displaying specific habitat characteristics that are unique to the study area. Our observations indicate that these exceptional caves possess habitat and landscape features that promote subterranean diversity, such as large dimensions, high humidity, and location within forested areas. Importantly, these caves are often geographically distant from one another. This highlights the fact that individual “supercaves” have the potential to accommodate troglobitic species in a given region of the study area at comparable or even higher levels of diversity compared to groups composed of multiple cavities. These findings suggest that a mixed conservation approach is necessary for the sustainable management of ferruginous geosystems, incorporating both individual prioritization of “supercaves” and collective prioritization of other caves that exert significant influence over singular caves. This concept aligns with the idea proposed by Ovaskainen (2002), which suggests that a large number of small patches may be optimal for long-term species persistence, as species ranges increase with the number of patches. Therefore, the most effective strategy for preserving subterranean communities in iron ore landscapes would involve the creation of a network of scattered reserves, as observed in other ecosystems (e.g., Baz & Garcia-Boyero, 1996; Mo et al., 2019; Liu, Li & Lv, 2019).

We also observed that caves with limited representation in terms of habitat and landscape characteristics, as well as a low number of troglobitic species, can serve as connectors between groups of caves. However, the question arises as to whether these caves should be considered for conservation within a framework of sustainable use. Contrary to what may be expected, this situation does not align with the concept of “stepping stones” (Moilanen, 2011), as the species are dispersed throughout the entire ferruginous matrix. Instead, these caves exert simultaneous influence on communities from different groups. Therefore, it is crucial to consider their conservation value when there are unique caves among the communities they influence.

Conclusion

In conclusion, it is crucial to develop protocols for the protection of subterranean biodiversity that encompass not only the caves themselves but also the surrounding landscape, with a particular focus on troglobitic species. Our graph-based approach enables environmental management at both group and individual cave levels. By applying this approach, we have elucidated the connectivity patterns, identified distinct groups of caves, highlighted the importance of preserving singular or unique caves, and identified caves that exert influence on these unique habitats. This methodological advancement is essential for the formulation of effective conservation policies aimed at safeguarding speleological heritage in areas facing ongoing economic pressures (Ferreira et al., 2022).

Supplemental Information

Supplemental Information 1 Datasets and R scripts.

Click here for additional data file.

Supplemental Information 2 Environmental variables measured in the external landscape and caves of Serra do Tarzan.

The type, unit, concept (description), code and method applied to the obtained data for each variable are informed.

Click here for additional data file.

Supplemental Information 3 Description of planimetric patterns applied to ferruginous caves (adapted from Calux, Cassimiro & Salgado, 2019).

Click here for additional data file.

Supplemental Information 4 Characterization of small-scale morphological features adapted to iron-ore caves according to Lundberg (2012).

Click here for additional data file.

Supplemental Information 5 List and taxonomic classification of species selected for the calculation of phylogenetic distance.

Click here for additional data file.

Supplemental Information 6 Summary of landscape and cave variables sampled at Serra do Tarzan.

The variable’s names, types, units and concepts (description) are in Table S1 and codebook.

Click here for additional data file.

Supplemental Information 7 Metrics used in graphs to defined conservation priorities.

For each cave, information is provided on the number of troglobitic species, connections established to other cavities (Degree), the uniqueness in relation to its troglobitic fauna and habitat specificity (Singularity), and the intermediation centrality measure (Betweenness), which indicates the potential of the cave as “bridges” between distant regions of the graph.

Click here for additional data file.

Supplemental Information 8 Troglobitic species sampled at Serra do Tarzan.

A) Pyrgodesmidae gen.1 sp.2 (Polydesmida: Pyrgodesmidae). B) Glomeridesmus sp.1 (Glomeridesmida: Glomeridesmidae). C) Pseudosinella sp.1 (Collembola: Lepidocyrtidae). D) Hyalella sp.2 (Amphipoda: Hyalellidae). E) Laelapidae sp.1 (Mesostigmata: Laelapidae). F) Trichorhina sp.1 (Isopoda: Platyarthridae). G) Circoniscus carajasensis (Isopoda: Scleropactidae). H) cf. Bogidiella sp.1 (Amphipoda: Bogidiellidae). I) Trogolaphysa sp.2 (Collembola: Entomobryidae).

Click here for additional data file.

Supplemental Information 9 Principal Coordinates Analysis (PCoA) according to the similarity approach (species composition or phylogenetic distance) applied to all the species or only troglobites.

We carried out PCoA based on the Bray-Curtis Index for species composition and Euclidean distance for phylogenetic relationships.

Click here for additional data file.

Supplemental Information 10 Phylogenetic relationship based on the distance between the taxonomic levels.

A) Considering all the cave Community (represented by 411 taxa identified at least at generic level, widely distributed in the study area). B) Only troglobites. Species groupings at class level are highlighted in the clusters.

Click here for additional data file.

Supplemental Information 11 Connections between pairs of caves (CPC) according to the connection weight (intensity of the link between pairs of caves).

A) Weight 2 (gray), B) Weight 3 (yellow), and C) Weight 4 (red). The vertex size expresses the number of troglobitic species sampled in the cavity.

Click here for additional data file.

Supplemental Information 12 Interactive 3D graph representing the subterranean network among caves in Serra do Tarzan based on the similarity of the troglobitic fauna (CPC = 0.214).

The vertex size expresses the number of troglobitic species sampled in the cave. Edge width and color represent the connection strength between pairs of caves: weight 2 (grey), weight 3 (yellow) and weight 4 (red), respectively.

Click here for additional data file.

We thank the taxonomists who helped with fauna identification: Dr. Antonio Brescovit (Araneae), Dr. Douglas Zeppelini (Collembola), Dra. Rafaela B. Pereira (Isopoda and Amphipoda), Dr. Leopoldo Bernardi (Acari), Msc. Luiz Iniesta (Diplopoda), Msc. Rodrigo Bouzan (Diplopoda), Dr. Ludson Azara (Opiliones), Dr. Amazonas Chagas Jr (Chilopoda), Msc. Victor Calvanese (Geophilomorpha and Lithobiomorpha), Dr. Ricardo Pinto da Rocha (Schizomida), Msc. Ana Vasconcelos (Amblypygi), Dra. Maysa Rezende Souza (Palpigradi), Dr. Angélico Asenjo (Coleoptera), Dra. Thaís Pellegrini (Coleoptera), Dr. Michel Valim (Diptera) and Msc. Luciana Falci (Oligochaeta). Special thanks to João Paulo Alves, Fagner Márcio Batista, Msc. Silvia Helena S. Torres, Thaís Pereira, Msc. Marta Letícia Kerkhoff, Diogo Chechia, Danilo Bebiano, Dra. Lisiane Zanella, Renato Freitas, Elbert Moreira, Daniel Miori, Acauã Rodrigues, Rodolfo Hass and Gislene Rios for helping us with fieldwork. Thami Gomes Oliveira and Juliano Belchior for the logistic organization. Matheus Brajão Mescolotti, Matheus Barreto Fernandes and Hemerson Gomes for the health and safety guidelines during the activities. Dr. Paulo Pompeu for his comments during the statistical analyses. We acknowledge the comments from Dr. Alberto Sendra Mocholí and two anonymous reviewers, which greatly enhanced our article.

Additional Information and Declarations

Competing Interests

Author Contributions

Field Study Permissions

Data Availability

Marcus Paulo Alves de Oliveira is employed by BioEspeleo Consultoria Ambiental.

Marcus P. A. Oliveira conceived and designed the experiments, performed the experiments, analyzed the data, prepared figures and/or tables, authored or reviewed drafts of the article, and approved the final draft.

Rodrigo L. Ferreira conceived and designed the experiments, performed the experiments, authored or reviewed drafts of the article, and approved the final draft.

The following information was supplied relating to field study approvals (i.e., approving body and any reference numbers):

Permission was given by Instituto Chico Mendes de Conservação da Biodiversidade (ICMBio), N° 83/2016, regarding the protocol N° 36/2016.

The following information was supplied regarding data availability:

The datasets and R scripts are available in the Supplemental Files.

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
