# Peer review of "Extending beyond individual caves: a graph theory approach broadening conservation priorities in Amazon iron ore caves"

_PeerJ, doi:10.7717/peerj.16877_

## Round 0.1 · original submission · Minor Revisions

The reviewers all very much liked your article, but each has suggested some different minor clarifications you can make. Please attend to these before sending the manuscript back.

Reviewer 1 ·

Basic reporting

This is an interesting manuscript, and mostly well written, manuscript.

The English needs occasional re-working to improve clarity. Some examples are line 25 “we assessed the predictors influencing the similarity among cave communities and proposed the use of this parameter as a proxy for subterranean connectivity”, line 32 “even in clusters densely interconnected by similarity, certain caves stood out for sheltering an exclusive fauna and/or presenting habitat specificity”, line 98 “we predict that strategies aimed at safeguarding clusters (groups of caves) will overlap individual preservation efforts (single cave)”, and line 110 “twice along 2016”.

I think that the Methods section provides insufficient information about how all data were collected. This is compounded by the lack of general description in the Results section. A small summary of the range of richness and environmental parameters within caves etc would save a confused reader from having to chase the supplementary data.

Experimental design

The research questions are clearly defined and the manuscript explores an approach to selecting sites for subterranean fauna conservation.

Work is performed to a high standard but I have some questions about relative humidity in graphs and analyses.

There is a need to briefly describe methods of collecting all data.

Validity of the findings

The manuscript provides interesting results and conclusions are well stated. In terms of conservation planning and the key issues associated with using results of the analysis, different people might see different priorities but the investigation has been undertaken in a thorough way. That may seem like a negative comment but it is actually intended as a positive one. The manuscript has caused me to think hard about site selection issue, selection of sites or lanndscapes for subterranean fauna conservation, and the importance or relevance of epigean fauna in the process.

The conclusions reflect the research questions asked and there is no speculation beyong directly interpreting results of the study.

Additional comments

This is an interesting manuscript, and mostly well written manuscript, but some of the relative humidit information confueses me and I think that the Methods section provides insufficient information about how all data were collected. This is compounded by the lack of general description in the Results section. A small summary of the range of richness and environmental parameters within caves etc would save a confused reader from having to chase the supplementary data.

I was surprised that only 1.3% of fauna is troglobitic (line 299). This may suggest troglobites have minimal ecological role in the caves and that in terms of the Amazonian biodiversity resource they are not an important component. Is the value of troglobites more in the evolutionary history of the individual troglobitic species than cave ecology?

While the manuscript in its current form is interesting, the preponderance of non-troglobites in the cave community perhaps means conservation planning should take at least some account of the occurrence of the recorded non-troglobite species outside caves as well as in caves. What unique role do caves have as habitat for most of the non-troglobite species using them?

Minor comments
The English needs occasional re-working to improve clarity. Some examples are line 25 “we assessed the predictors influencing the similarity among cave communities and proposed the use of this parameter as a proxy for subterranean connectivity”, line 32 “even in clusters densely interconnected by similarity, certain caves stood out for sheltering an exclusive fauna and/or presenting habitat specificity”, line 98 “we predict that strategies aimed at safeguarding clusters (groups of caves) will overlap individual preservation efforts (single cave)”, and line 110 “twice along 2016”.

In line 85 it is indicated that the aim is to preserve speleological heritage but the five questions this generates on line 87 are arguably concerned only with cave fauna (even the cave connectivity question). Shouldn’t the explicit aim be to protect/conserve cave fauna?

Line 96 – is the likelihood of migration high or low and over what distance is it expected?

Line 105 – does nothing happen in May?

Line 111 – measured rather than collected.

Line 119 – do not understand what is meant by “identified supra-specific taxa as morpho-species”. Morpho-species are usually identifications based on morphology that will separate most species but lump cryptic species, whereas supra-specific taxa are genus, family, order. More explanation is needed. Oliver and Beatttie (1996) did as described above (note misspelling in citation of this paper).

Line 121 – mean troglobites were recognised …?

213 – troglobitic.

217 – it can indicate either migration, or persistence of a surface population that moved underground across the range of caves in which the species is found, or possibly very occasional surface movement on wet nights.

Lines 302 - this prediction is not accurate to less than 1m, better to say 700 m (same for other distances and other numbers – too many non-significant digits).

Relative humidity information confused me. I don’t understand 3.999% RH in “relative air humidity with similar values (MaxH ± 3.999% RH) (Fig. 3C)” (line 331). Table S5 shows MaxH varies between 88 and 99.

Again, looking at the supplementary data I am confused why the following applies “relative humidity (MaxH ± 6,000% RH) … (Fig. 3D) (line 333). In Fig. 3D the relevant graph is labelled Min. humidity. Note comma instead of full stop.

Figure 3 – re humidity, is %UR Portuguese for %RH?

·

Basic reporting

It is a well done article, made correctly, for my part I have not doubt about the publication. Nevertheless, I have some suggestions to the authors. Most of the fauna or no cave-adapted and their stadistical weight are to mauch for only nine species; in the fuure I suggest to define the tipology of the cave-cammunities: entrance, twilling, deep zones.

Experimental design

I am impressed about all the analysis made in this article. These are very robust. I have only some suggestion for future works, that is the use of the morphology of the caves: entrance size, type of cave, mesurements of the cave zones mostly if it is used non cave-adapted fauna.

Validity of the findings

All is ok.

Reviewer 3 ·

Basic reporting

I think the text is very well written, precise and clear, therefore, the English language must not be improved. Sufficient literature references were provided. The article is professionally written, following the article structure required. Photos, table and supplementary materials are clear. The work is very interesting and useful for research community, highlighting fundamental points for subterranean biodiversity conservation.

Experimental design

The research is original and follows the aim and scope of the journal. Rigorous investigation were performed with high standard. Methods are described well, however, I suggest to improve some informations in Materials and Method (see athached file), especially in "Sampling cave fauna and variables" . I suggest also to improve information on the study area.
Moreover, I think it would be more correct and important to call the underground animals, troglobes and not, as hypogean species, because they are dependent on the subterranean environment, and not dependent on the cave environment only. These animals live not only in caves, but in the whole system of fractures and cracks in the ground. Caves are just larger cracks where humans can enter, but these animals also live in smaller spaces.

Validity of the findings

The research is novel and original, and could have a high impact on the research community and for habitat and species conservations. Statistical analysis are robust. Conclusion are well stated.

Additional comments

I suggest the admission of this interesting work after the small reviews recommended.

Annotated reviews are not available for download in order to protect the identity of reviewers who chose to remain anonymous.

---

## Round 0.2 · accepted · Accept

Thank you for your replies to the reviewers. The manuscript is now suitable for publication.